

# Patchiness of forest landscape can predict species distribution better than abundance: the case of a forest-dwelling passerine, the short-toed treecreeper, in central Italy

Marco Basile[1,2,3], Francesco Valerio[4], Rosario Balestrieri[1,2], Mario Posillico[1,5], Rodolfo Bucci[5], Tiziana Altea[5], Bruno De Cinti[1] and Giorgio Matteucci[6]

[1] Istituto di Biologia Agroambientale e Forestale, Consiglio Nazionale delle Ricerche, Monterotondo Scalo, Italy
[2] Coordinamento MItO2000, Parma, Italy
[3] Chair of Wildlife Ecology and Management, Albert-Ludwigs-Universität Freiburg, Freiburg, Germany
[4] CIBIO/InBIO-UE—Research Center in Biodiversity and Genetic Resources, Pole of Évora Applied Population and Community Ecology Laboratory, University of Évora UBC—Conservation Biology Lab, Department of Biology, Évora, Portugal
[5] Ufficio Territoriale Biodiversità di Castel di Sangro-Centro Ricerche Ambienti Montani, Corpo Forestale dello Stato, Castel di Sangro, Italy
[6] Istituto per i Sistemi Agricoli e Forestali del Mediterraneo, Consiglio Nazionale delle Ricerche, Ercolano (Na), Italy

Corresponding author
Marco Basile, marcob.nat@gmail.com

## ABSTRACT

Environmental heterogeneity affects not only the distribution of a species but also its local abundance. High heterogeneity due to habitat alteration and fragmentation can influence the realized niche of a species, lowering habitat suitability as well as reducing local abundance. We investigate whether a relationship exists between habitat suitability and abundance and whether both are affected by fragmentation. Our aim was to assess the predictive power of such a relationship to derive advice for environmental management. As a model species we used a forest specialist, the short-toed treecreeper (Family: Certhiidae; *Certhia brachydactyla* Brehm, 1820), and sampled it in central Italy. Species distribution was modelled as a function of forest structure, productivity and fragmentation, while abundance was directly estimated in two central Italian forest stands. Different algorithms were implemented to model species distribution, employing 170 occurrence points provided mostly by the MITO2000 database: an artificial neural network, classification tree analysis, flexible discriminant analysis, generalized boosting models, generalized linear models, multivariate additive regression splines, maximum entropy and random forests. Abundance was estimated also considering detectability, through N-mixture models. Differences between forest stands in both abundance and habitat suitability were assessed as well as the existence of a relationship. Simpler algorithms resulted in higher goodness of fit than complex ones. Fragmentation was highly influential in determining potential distribution. Local abundance and habitat suitability differed significantly between the two forest stands, which were also significantly different in the degree of fragmentation. Regression showed that suitability has a weak significant effect in explaining increasing value of abundance. In particular, local abundances varied both at low and high suitability

values. The study lends support to the concept that the degree of fragmentation can contribute to alter not only the suitability of an area for a species, but also its abundance. Even if the relationship between suitability and abundance can be used as an early warning of habitat deterioration, its weak predictive power needs further research. However, we define relationships between a species and some landscape features (i.e., fragmentation, extensive rejuvenation of forests and tree plantations) which could be easily controlled by appropriate forest management planning to enhance environmental suitability, at least in an area possessing high conservation and biodiversity values.

# INTRODUCTION

In recent years, considerable research effort has been involved in studying the influence of landscape patterns on biodiversity, triggered by the wide availability of biological data, as well as by the development of sophisticated species distribution models (SDMs), capable of predicting the presence of a species as a function of environmental variables (*Elith & Leathwick, 2009*). The reliability of SDMs is based on the quality of occurrence data and the use of environmental predictors linked to species occurrence (*Austin, 2007*). For instance, presence data collected through nationwide standardised monitoring programmes provide enormous advantages in using SDMs, due to the creation of large databases (*Elith & Leathwick, 2009*), hosting large amounts of occurrences and covering a wide, biologically significant area. Appropriate environmental predictors are those supposed to best describe the set of abiotic and biotic conditions affecting species occurrence, i.e., those characterising the species ecological niche (*sensu* Hutchinson; *Hutchinson, 1957*; *Holt, 2009*). Indeed, large-scale species distribution modelling can be useful for addressing species-habitat relationships at multiple spatial scales in order to understand the spatial variability in habitat selection (*Farashi, Kaboli & Karami, 2013*; *Chefaoui et al., 2015*; *Morand et al., 2015*). Also, considering the spatial heterogeneity in the environment has become essential in many studies regarding reproduction, meta-population dynamics, gene flow, dispersal and connectivity (*Bender, Tischendorf & Fahrig, 2003*; *Wang et al., 2008*; *Ryberg et al., 2013*). Recent studies have addressed this issue to propose alternative conservation strategies (*Nixon et al., 2014*), to monitor landscape change (*Darvishi, Fakheran & Soffianian, 2015*) and to give insight into the distribution of native and non-native species (*Kumar, Stohlgren & Chong, 2006*). Moreover, spatial patterns are considered major drivers of many ecosystem processes (*Uuemaa, Mander & Marja, 2013*).

Although landscape heterogeneity may promote biodiversity due to the increase in habitat types (i.e., spatial heterogeneity) (*Wiens, 1976*; *Loehle et al., 2005*; *Schindler et al., 2013*), a highly diverse landscape arising from anthropogenic fragmentation may result in the loss of natural habitats and specialist species, which frequently require large patches of relatively unaltered habitat (e.g., extensive areas of well-preserved forests) (*Marvier, Kareiva & Neubert, 2004*). Therefore, fragmentation can sometimes produce a simplification of

the biological community, or biotic homogenisation across the landscape (*McKinney & Lockwood, 1999*). Such a consequence derives from the loss of unique habitats, which are not replaceable in the short term (*Fahrig, 2003*).

Species abundance is also influenced by spatial variability, being affected by spatial gradients in the environmental parameters that form the environmental niche (*Martínez-Meyer et al., 2013*). Optimal conditions can be found where the environmental parameters are close to the centroid of the Hutchinsonian niche (*Hutchinson, 1957*). Hence, environmental variability can influence both the presence and abundance of a species. Indeed, the decrease in abundance could warn about a species decline in population and/or range extent earlier than a decrease in environmental suitability. In fact, abundance could also be low in highly suitable regions, in response to local limiting factors (*VanDerWal et al., 2009*).

The aim of our study was to investigate the relationship between environmental suitability and abundance of a species, in response to fragmentation. However, true environmental suitability can be expressed only by the whole set of environmental predictors and the local conditions that can influence movements and interaction (*Grinnell, 1917*) and the persistence of those conditions itself (*Jackson & Overpeck, 2000*). Such an approach may be unfeasible, as in our case. Therefore, we refer to a restricted set of factors influencing local or regional environmental suitability, i.e., some environmental predictors, which are supposed to be related to the probability of occurrence, and concern habitat suitability (HS) (*Franklin, 2009*). Among those habitats that can be highly modified by human activities, our research focused on forests, where unsustainable timber harvest can result in a patchy landscape and alter the habitat, adversely affecting forest biodiversity (*Donald et al., 1998*; *Penman, Mahony & Lemckert, 2005*; *Craig, 2007*; *Bearer et al., 2008*; *Shifley et al., 2008*; *Czeszczewik et al., 2014*; *Calladine et al., 2015*; *Escobar et al., 2015*). Woody plants are key elements in shaping the distribution of several bird species *such as* birds (*MacArthur, Recher & Cody, 1966*; *Cody, 1985*). Landscape structures and the spatial arrangement of habitat patches can affect both the abundance and distribution of birds, acting as structural bio-modifiers (*Uuemaa, Mander & Marja, 2013*).

Therefore, we selected as a model species a forest specialist bird, the short-toed treecreeper (Family: Certhiidae; *Certhia brachydactyla* Brehm, 1820), and used landscape metrics as well as forest variables to characterise the forest landscape and weight habitat suitability. The short-toed treecreeper is considered a forest-dwelling passerine, and hence a forest specialist, being a secondary cavity nester (*Newton, 1994*). It is usually found in oak or mixed-deciduous forests (with prevalence of oak), where it nests inside small holes excavated by woodpeckers or left by dead branches (*Cramp, 1988*). It is a resident species, with very limited movements, usually restricted to post-juvenile dispersal (*Cramp, 1988*). Home range and territory size can also be very limited, sometimes less than 1 ha (*Cramp, 1988*). The global range extends through most of central/southern Europe, up to Turkey and the Caucasus, overlapping with *C. familiaris* in central Europe (*BirdLife International, 2016*). Thus, we assessed whether there was a relationship between local abundance and HS. We hypothesise that the realized niche can be altered by fragmentation, resulting in lower HS and abundance. If such a relationship emerges, we aim to estimate its predictive power and usefulness in forest management and in conservation policies. In addition, we

modelled HS with several algorithms and compared results, to assess whether different species distribution models (SDMs) follow the same pattern of response.

## METHODS

One of the main advantages offered by SDMs relies on the use of occurrence data collected with different methods (*Tsoar et al., 2007*). Therefore we used occurrence records from multiple sources, that spanned from year 2000 to 2013. We relied mainly on the MITO2000 database (*Monitoraggio Italiano Ornitologico*, Italian Ornithological Monitoring), an ongoing project which started in 2000 and operates at a country-wide level (*Fornasari et al., 2010*). The project uses point counts with unlimited radius (*Blondel, Ferry & Frochot, 1981*), sampling points being randomly selected within a 1 km² grid square in the region of interest. Point counts were carried out during a short time frame, from mid-May to mid-June. Occurrences of *C. brachydactyla* were also extracted from the databases of the National Forest Service (Ufficio Territoriale della Biodiversità, Castel di Sangro, AQ), and the LIFE+ ManFor C.BD project, which employed a sampling design similar to MITO2000, albeit at a smaller spatial scale (∼200–500 m). The spatial coverage of the occurrences was limited to the administrative boundaries of the regions of Lazio, Abruzzo and Molise, comprising 32,523 km², of which over one-third (12,309 km²) had forest cover (Fig. 1). The whole database was filtered from all the pseudo-replicated points that fell into the same 1 km² grid. The database was further cleared of all the occurrences that were located in unrealistic locations (i.e., non-forested areas), except for those <300 m away from the nearest forest patch, which were relocated to the nearest patch. Every occurrence was georeferenced with GPS. Hence, for our purposes, the error in location was assumed to be the same across the three datasets. The final database consisted of 170 occurrence points of *C. brachydactyla* (Table 1), of which 119 were supplied by the MITO2000 database, exceeding the recommended minimum sample size (*Wisz et al., 2008*).

### Species distribution models

The SDMs were implemented using five environmental predictors, correlated with forest structure, productivity and the degree of fragmentation, at a spatial resolution of 30 m. First, a habitat type map, consisting of 12 classes, was created from the regional forest maps (*Marchetti, Chiavetta & Santopuoli, 2009*; *Garfí & Marchetti, 2011*; *Open Data Lazio, 2012*), aggregating all of the non-forest habitat and distinguishing 11 forest types (Table 1). Three landscape metrics were then calculated from the habitat map, using FRAGSTATS v. 4 software (*McGarigal, Cushman & Ene, 2012*): (1) *Diversity* (H'), a measure of patch type diversity within the landscape (*Shannon & Wiener, 1949*); (2) *edge density* (ED) which expresses the density (m ha⁻¹) of boundaries; (3) the *aggregation index* (AI) which measures the degree of aggregation between forest patches (*He, Dezonia & Mladenoff, 2000*). The first two metrics were implemented using a moving window of 1,000 m, as they can show little variability among different spatial scales and we were interested in landscape features, avoiding the influence of small patches (*Uuemaa, Roosaare & Mander, 2005*). By contrast, for AI a 300 m moving window was used, as we were interested, in this case, in controlling how small patches aggregate across the landscape, according also to the home
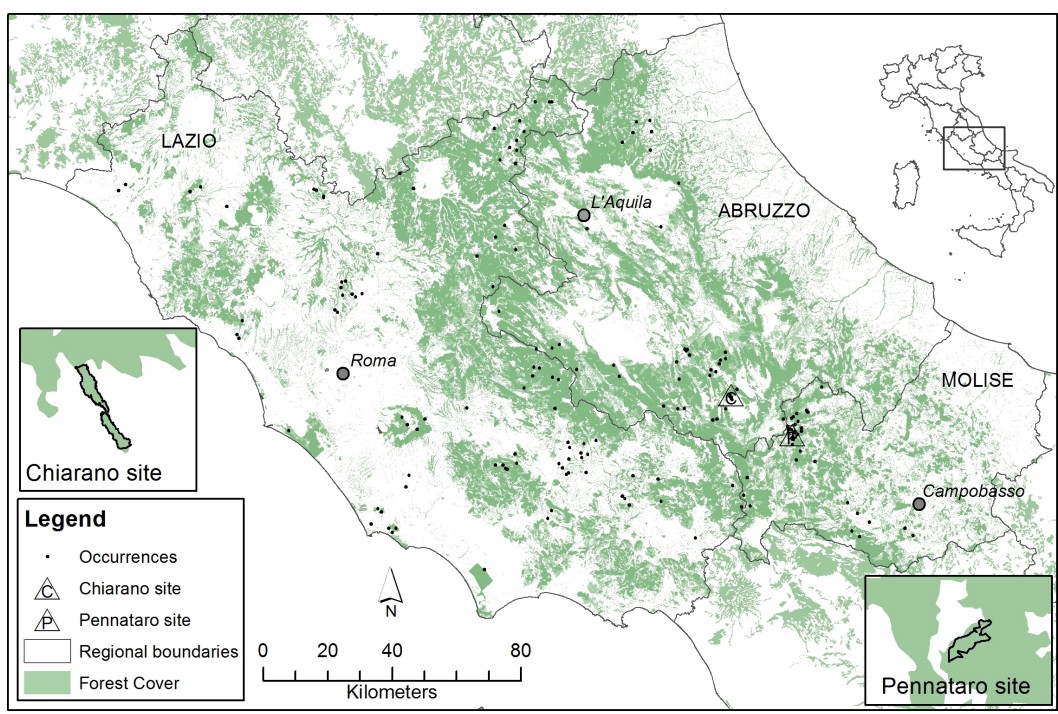

**Figure 1** **Treecreeper's occurrences used to build the distribution models.** The study area is located in central-southern Italy, within Abruzzo, Lazio and Molise regions.

**Table 1** **Surface of the habitat types included in the analysis within the study area (Abruzzo, Lazione and Molise regions, central Italy) and number of short-toed treecreeper's occurrences.**

| Forests and tree plantations habitat types | Area (km²) | N° of treecreeper's occurrences |
|---|---|---|
| Holm oak (*Quercus ilex*) | 511.9 | 8 |
| Downy oak (*Q. pubescens*) | 1986.3 | 13 |
| Turkey oak (*Q. cerris*) | 2412.3 | 51 |
| *Orno-ostryetum* (mixed deciduous woodland with prevailing *Fraxinus ornus* and *Ostrya carpinifolia*) | 1342.4 | 20 |
| Chestnut (*Castanea sativa*) | 628.1 | 10 |
| *Tilio-Acerion* | 0.12 | 0 |
| Beech (*Fagus sylvatica*) | 2360.4 | 40 |
| *Salix* sp. and *Populus* sp. riparian woodlands and poplar plantations | 536.5 | 12 |
| Tree plantations and bushes | 649.7 | 8 |
| Conifer (both natural and reforestation) | 545 | 4 |
| Shrubland and maquis | 1313.1 | 4 |
| Non forest | 20129.3 | 0 |

range and territory size of the short-toed treecreeper (*Cramp, 1988*). Accordingly, we chose to use the normalized difference vegetation index (NDVI) as a proxy of forest cover and structure, integrating it into the modelling framework. The NDVI is highly correlated with the leaf area index and the net primary productivity (*Myneni et al., 1995*; *Pettorelli et al., 2005*; *Lee et al., 2006*) and was calculated from Landsat 8 multispectral images, with 30 m

spatial resolution. NDVI was computed over a mosaic of five images with cloud cover <10%, collected between July and August 2013, which had undergone the atmospheric correction procedure. Finally, altitude was integrated through a digital elevation model (DEM) provided by the National Institute for Environmental Protection and Research (ISPRA), available at http://www.sinanet.isprambiente.it/it.

Spatial autocorrelations of the environmental predictors within occurrence points were tested through a Mantel test in order to detect any spatial autocorrelations among occurrences (Fig. S1). Analyses were carried out with the R package 'ecospat' (Broennimann, Di Cola & Guisan, 2016).

Among the eight selected algorithms, the maximum entropy (ME) used presence-only points in combination with background samples, using only quadratic and hinge features to avoid overfitting (Phillips, Anderson & Schapire, 2006; Elith et al., 2011). The other algorithms, which were supplied with pseudo-absences and true absences, were: an artificial neural network (ANN; Segurado & Araujo, 2004), classification tree analyses (CTA; Breiman et al., 1984; De'ath, 2002), flexible discriminant analyses (FDA; Hastie, Tibshirani & Buja, 1994), generalized boosting model (GBM; Friedman, 2001), generalized linear model (GLM; McCullagh & Nelder, 1989), multivariate additive regression spline (MARS; Moisen & Frescino, 2002) and random forest (RF; Breiman, 2001) (Table 2). Ten thousand absence points were sampled in the environmental background (Elith et al., 2006), comprising 975 points of actual absence derived from the MITO2000 database and 9025 pseudo-absences, randomly selected within the area where the logistic output of ME was less than 0.2 (Chefaoui & Lobo, 2008; Wisz & Guisan, 2009), representing an adequate number of pseudo-absences (Barbet-Massin et al., 2012). SDMs were trained using 70% of randomly selected occurrences, while the remaining 30% were used for testing; the procedure was iterated 30 times (except for ME with 50 iterations) (further details are provided in Table 2). The area under the curve (AUC) of the receiving operating characteristic (Hanley & McNeil, 1982) was used to evaluate the predictive power of the SDMs. To improve the readability of SDM outputs, sensitivity (i.e., the proportions of correct positive prediction) and specificity (i.e., the proportion of correct negative prediction) and the true skill statistic (TSS) were also reported (Allouche, Tsoar & Kadmon, 2006; Lobo, Jiménez-Valverde & Real, 2008). The importance of each environmental predictor was calculated following Thuiller et al. (2009). Analyses were carried out with the software MaxEnt (Phillips, Anderson & Schapire, 2006) and the biomod2 package integrated in R (Thuiller et al., 2009; R Development Core Team, 2015; Thuiller, Georges & Engler, 2015).

## Abundance estimation

Abundance was estimated in two forest stands used as test sites of the LIFE+ ManFor C.BD: Bosco Pennataro Regional Forest and Chiarano-Sparvera Regional Forest. Bosco Pennataro (BP, 41°44′N, 14°11′E, 1,000 m a.s.l.) consists of a multi-layered high forest stand dominated by turkey oak (Quercus cerris). Chiarano-Sparvera (CS, 41°51′N, 13°57′E, 1,700 m a.s.l.) is a pure beech (Fagus sylvatica) forest, in transition from coppice to high forest. Following a systematic design, 27 and 23 sampling points, 125.5 m (±19.7 sd) away from one another, were selected in BP and CS, respectively. Surveys were carried out from
**Table 2** Settings used for species distribution modelling and resulted AUC (area under the curve of the receiving operator characteristic), sensitivity, specificity and TSS (true skills statistic).

| Full name | Acronym | Pseudo-absences | Parameters | AUC | Sensitivity | Specificity | TSS |
|---|---|---|---|---|---|---|---|
| Artificial neural network | ANN | 10,000 | 5-fold cross validation | 0.949 | 92.045 | 89.689 | 0.771 |
| Classification tree analyses | CTA | 10,000 | 5-fold cross validation | 0.918 | 85.795 | 93.839 | 0.792 |
| Flexible discriminant analyses | FDA | 10,000 | Default with MARS to increase predictive power | 0.894 | 82.955 | 93.849 | 0.768 |
| Generalized boosting model | GBM | 10,000 | 5,000 maximum trees, 5 interaction and 10-fold cross validation | 0.961 | 93.75 | 94.529 | 0.842 |
| Generalized linear model | GLM | 10,000 | AIC-based stepwise model selection | 0.959 | 93.182 | 91.159 | 0.835 |
| Multivariate additive regression splines | MARS | 10,000 | Spline knots are determined automatically | 0.913 | 89.205 | 89.129 | 0.782 |
| Maximum entropy | ME | No; 10,000 background points | 1,000 bootstrap iterations | 0.929 | – | – | – |
| Random forest | RF | 10,000 | 750 trees, 10-fold cross validation | 1 | 100 | 99.98 | 1 |

May to June (2012 in CS; 2013 in BP) from sunrise till 11:00 a.m. At every point, each individual detected by aural/visual cues during a five-minute count was recorded. Each point was visited two to six times (average = 3.4; total = 177).

Local abundance was estimated with N-mixture models (*Royle, 2004b*). This approach considers local abundance (i.e., abundance estimated in each sampling point) as an independent random point process (*Royle, 2004a*). Two separate models were built for BP and CS, respectively: with and without detectability variation among occasions. Model fit and overdispersion (also called c-hat) was tested through a Pearson $\chi^2$ goodness-of-fit test, with 1,000 bootstrap resampling (*MacKenzie & Bailey, 2004*). Model selection proceeded through Akaike's Information Criterion, which assigns scores both to the likelihood of the model and the number of parameters included (*Burnham & Anderson, 2002*). Spatial dependence of estimates was assessed with the Moran test and index calculation (*Moran, 1950*). Analyses were carried out using the packages *unmarked* (*Fiske & Chandler, 2011*), *AICmodavg* (*Mazerolle, 2015*) and *spdep* (*Bivand & Piras, 2015*) implemented in R (*R Development Core Team, 2015*).

## Statistical analyses

Local abundances (i.e., the abundance at every sampling point) in BP and CS were tested for differences with an *F*-test, followed by a *t*-test. Habitat suitability values, defined as the SDM outputs, were then extracted from a discrete area surrounding every abundance point. Width of the area in question was proportional to local abundance and was derived by

transforming the estimated population size (i.e., the sum of local abundances) into densities (ind./ha): the area of interest for density transformation was given by the minimum convex polygon among the sampling points. The difference between BP and CS environmental suitability values was tested by an $F$-test and a $t$-test. The landscape metric values were also tested for difference with the same methods.

The relationship between abundance and environmental suitability can form a triangular envelope, where increasing values of environmental suitability are matched by increasing values of the maximum abundance, not just the mean abundance (*VanDerWal et al., 2009*). Therefore, quantile regression can best provide the opportunity to explore the relation between environmental suitability and the upper quantiles of the abundance (*Cade, Noon & Flather, 2005*). The triangular envelope can predict maximum abundance, given a suitability value, due to the increase in the slope of regressions of upper quantiles, while intercepts remain similar (*VanDerWal et al., 2009*). However, two factors can mask the results: first, random variation at every point also due to local limiting factors that are not feasible to model; secondly, the spatial structure of the data, that can generate autocorrelation. Therefore, quantile mixed regressions were implemented to model the abundance as a function of HS values of every SDM, with a null random term and a grouping factor identifying the two locations. The random effect is estimated through best linear prediction (*Geraci & Bottai, 2013*). Model fit was assessed for every quantile through comparison of AIC scores with the null model of the corresponding quantile (*Burnham & Anderson, 2002*). Statistical analysis was carried out with the *lqmm* package (*Geraci, 2014*) in R (*R Development Core Team, 2015*).

## RESULTS

Each SDM showed an AUC > 0.9, except for FDA (Table 2). Among them, RF ranked the highest value (AUC = 1). However, the geographical projections of the SDMs proved dissimilar (see Fig. S2). The importance of each environmental predictor had the same pattern for every algorithm, with forest type and NDVI proving the most important (Fig. 2). The importance of the three landscape metrics (H, AI, ED) indicates that the spatial configuration of landscape structures exerts a major influence on potential distribution.

Abundance models that performed best in both study areas were those in which detectability was invariant between sessions. Detectability was 0.34 ($\pm 0.11$ SE) in Bosco Pennataro and 0.21 ($\pm 0.27$ SE) in Chiarano Sparvera. Local abundances significantly differed between the two areas ($F = 0.77$, $p = 0.53$; $t = -3.57$, $p < 0.001$), and mean estimates were 1.54 ($\pm 0.52$ SE) in BP and 0.86 ($\pm 1$ SE) individuals/point in CS. Both models returned a good fit, with no overdispersion (BP: $\chi^2 = 64.3$, $p = 0.997$, c-hat = 0.687; CS: $\chi^2 = 52.5$, $p = 0.391$, c-hat = 1). Estimates did not show spatial autocorrelation in the two forest stands, obtaining a Moran I of 0.11 ($p = 0.14$) and $-0.26$ ($p = 0.92$) for BP and CS, respectively.

Habitat suitability also proved different between BP and CS, for every SDM (Table 3), and HS was higher in BP. In parallel, the values of landscape metrics significantly differed between the two forest stands except for edge density (Table 3). Specifically, Bosco Pennataro landscape structure resulted in larger and less scattered patches (AI = 98),

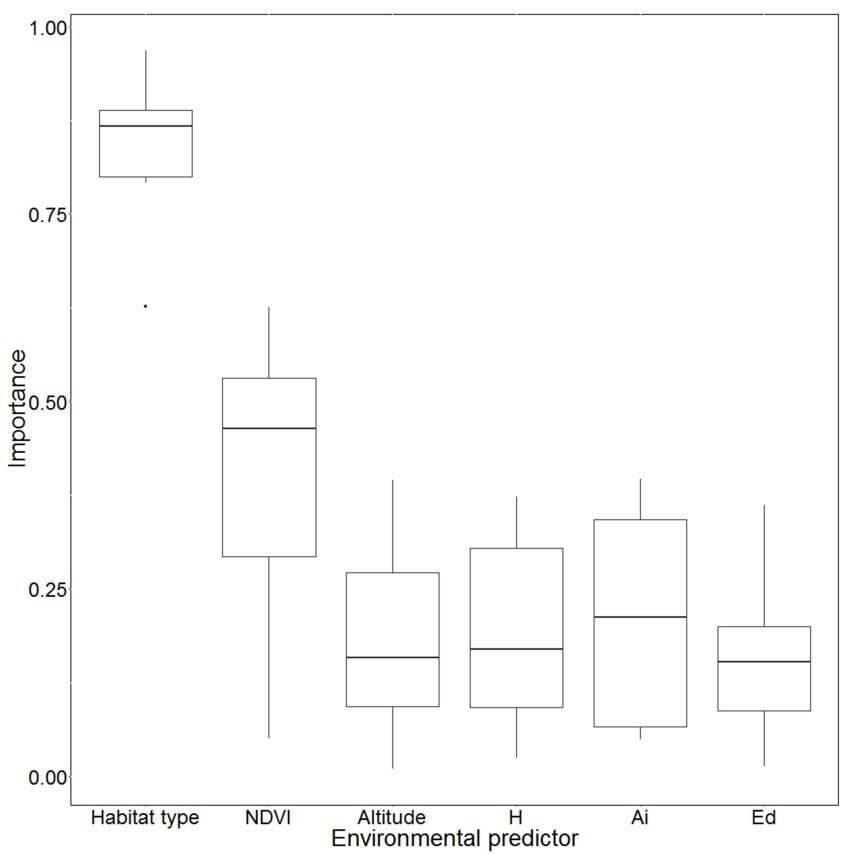

**Figure 2 Variable importance based on different Species Distribution Models (SDMs).** NDVI, Normalized difference vegetation index; H', Shannon index computed on landscape patch type diversity; Ai, aggregation index of landscape patches; Ed, patches' edge density.

equally distributed among types (H' = 0.93), compared to CS (AI = 92.6; H' = 0.76). Hence, landscape metrics showed a more fragmented landscape in CS than in BP, as expected.

Quantile regression showed a positive relationship between abundance and HS (Fig. 3 and Fig. S3). No differences emerged for the regression slope of each quantile, while intercept values proved more variable. Moreover, the majority of slopes were not significant except for CTA, GBM and GLM (see Table S1), even if AIC comparison indicated that most of the quantiles performed better than the corresponding null model (Table 4).

## DISCUSSION

We examined the abundance and habitat suitability resulting from many algorithms for species distribution modelling (*Elith et al., 2006*; *Li & Wang, 2013*) of a forest-dwelling passerine in a region with different degrees of fragmentation. Although SDMs showed high AUCs, geographical projections varied quite substantially among algorithms, even if their explanatory variables followed the same pattern of importance scoring. Moreover, AUC computation for ME differs from the other algorithms, which made use of (pseudo) absences, being not comparable (*Yackulic et al., 2013*). Several studies that compared SDM

**Table 3** Test for differences of landscape metrics and environmental suitability between Bosco Pennataro and Chiarano-Sparvera, based on Species Distribution Models (SDMs).

|  | *F* | *P* | *t* | *p* |
|---|---|---|---|---|
| Metric |  |  |  |  |
| H' | 0.065 | 0.000 | 3.3342 | 0.0027 |
| Ed | 0.3583 | 0.0134 | −1.5038 | 0.1392 |
| Ai | 0.221 | 0.000 | 7.1504 | 0.000 |
| Model |  |  |  |  |
| ANN | 0.07 | 0.000 | −36 | 0.000 |
| CTA | 4.901 | 0.000 | −9.93 | 0.000 |
| FDA | 2433.4 | 0.000 | −8.06 | 0.000 |
| GBM | 1.137 | 0.748 | −10.91 | 0.000 |
| GLM | 2.996 | 0.008 | −2.949 | 0.002 |
| MARS | 14648 | 0.000 | −4.893 | 0.000 |
| ME | 46.35 | 0.000 | −4.682 | 0.000 |
| RF | 30.42 | 0.000 | −4.044 | 0.000 |

**Notes.**

H', Shannon index of patch type diversity; Ed, edge density; Ai, aggregation index; $F$, Fisher's test; $t$, $t$ test; $P$, $p$ value; model abbreviation are given in Table 2.

**Table 4** DeltaAIC between null model and suitability-dependant model, for the same quantile.

| Quantile | ANN | CTA | FDA | GBM | GLM | MARS | ME | RF |
|---|---|---|---|---|---|---|---|---|
| 0.5 | 0 | 0 | 0 | 0 | 0 | 0 | 0 | 0.50 |
| 0.55 | 0 | 0 | 0 | 0 | 0 | 0 | 0 | 1.20 |
| 0.6 | 0 | 0 | 0 | 0 | 0 | 0 | 0 | 1.39 |
| 0.65 | 0 | 0 | 0 | 0 | 2.88 | 0 | 0 | 1.86 |
| 0.7 | 0 | 0 | 0 | 0 | 0 | 0 | 0.17 | 0.93 |
| 0.75 | 0 | 0 | 0 | 0 | 0 | 0 | 0 | 0 |
| 0.8 | 0 | 0 | 0 | 0 | 4.67 | 23.93 | 0 | 0 |
| 0.85 | 25.74 | 0 | 0 | 0 | 2.17 | 7.79 | 6.29 | 9.14 |
| 0.9 | 0 | 31.93 | 1.15 | 26.85 | 0 | 23.91 | 2.98 | 3.07 |
| 0.95 | 44.78 | 16.49 | 3.38 | 0 | 15.94 | 6.83 | 4.35 | 0 |
| 0.975 | 32.65 | 0 | 0 | 0 | 0 | 31.32 | 0 | 0 |
| 0.99 | 0 | 0 | 0 | 0 | 0 | 0 | 0 | 1.69 |

outputs differed substantially (*Segurado & Araujo, 2004*; *Elith et al., 2006*; *Moisen et al., 2006*; *Meynard & Quinn, 2007*). Among those that based their comparison on AUC, ANN was favoured over CTA and GLM (*Segurado & Araujo, 2004*) and GBM and ME were favoured over MARS and GLM (*Elith et al., 2006*). GBM and GLM were preferred to CTA also by other authors (*Moisen et al., 2006*; *Meynard & Quinn, 2007*). What emerged from the literature is that complex models usually outperform simple models, especially when they involve specialist species (*Jiménez-Valverde, Lobo & Hortal, 2008*; *Li & Wang, 2013*). Indeed, our results are not totally concordant with findings elsewhere, simple models like GLM and GBM having scored higher AUC values. However, the use of AUC has been criticised by many authors even if there are currently no consensus methods to assess the

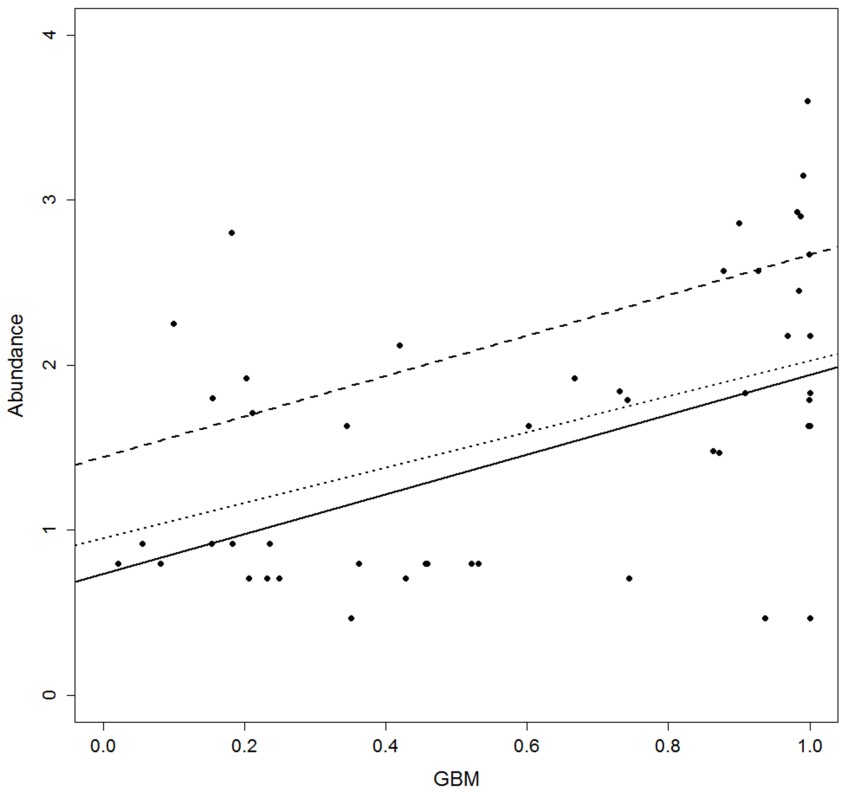

**Figure 3 Scatterplot of abundance versus habitat suitability (as predicted by the Generalised Boosting model, GBM).** Regression lines represent the fitted relationship at different quantiles. Quantiles: solid line = 0.5 quantile, slope = 0.37, $p < 0.5$; dashed line = 0.75, slope = 0.19, $p$ = n.s.; dotted line = 0.95, slope = 0.13, $p$ = n.s.

predictability of SDMs (*Austin, 2007*; *Lobo, Jiménez-Valverde & Real, 2008*). Specifically, the AUC does not consider the goodness of fit of a model and it is higher when more pseudo-absences in unsuitable localities are included in the model (*Lobo, Jiménez-Valverde & Real, 2008*). Nevertheless, its use is still widespread (*Elith & Graham, 2009*; *Barbet-Massin et al., 2012*). It should also be pointed out that, even if we used a large number of (pseudo) absences, we also employed a larger number of presence points than what is usually found in the literature (e.g., *Pearson et al., 2007*).

SDM output can usually be considered as a probability of occurrence, somewhat related to habitat suitability (*Franklin, 2009*). In the case of ME, this is achieved after logistic transformation (*Phillips & Dudík, 2008*). This approach has been criticised because of the frequent violation of two major assumptions: randomness of the samples and constant detectability among individuals (*Royle et al., 2012*; *Merow, Smith & Silander, 2013*). Indeed, the logistic output uses a rather subjective intercept of 0.5, which is valid, though its reliability is not proven (*Royle et al., 2012*). Use of pseudo-absence also needs caution, since the background in which sampling takes place has both suitable and unsuitable locations (*Pearce & Boyce, 2006*). However, we employed an analytical framework designed to reduce this source of bias. The randomness of the presence points is due to the use of occurrences coming from a standardised monitoring programme. For the same reasons, we

assume that the variability in detectability is reduced to the minimum, demonstrating this issue also in the two forests where we estimated abundance (i.e., BP and CS). Finally, our use of the logistic output of ME, as well as of its subjective intercept of 0.5, is based upon the consideration that 37% of the study area is covered in forest. Therefore, assuming an intercept of 0.5 does not seem too far from reality scenario. Indeed, ME has been proved to be one of the most reliable SDMs when only presence data are available (*Franklin, 2009*; *Merow, Smith & Silander, 2013*). Our use of ME, moreover, was functional to the selection of pseudo-absences, which were not selected within the entire region, but only in a restricted area considered unsuitable by ME. As a consequence, we also assume that our method of selecting pseudo-absence greatly reduced an eventual bias. At the very end, we considered SDM outputs as a habitat suitability index, which we could assume to be related to actual environmental suitability (*VanDerWal et al., 2009*; *Brambilla & Ficetola, 2012*).

For reliable modelling, it is necessary to use ecologically relevant environmental predictors (*Austin, 2007*), even if it is not possible to include every environmental variable thought to affect the distribution of a species (*Elith & Leathwick, 2009*). We based the choice of environmental variables on both the known species-habitat relationships and on the possibility of obtaining relevant information to steer management, relying on forest type, structure, productivity and fragmentation. Forest type and NDVI proved the most important variables in predicting the distribution of the short-toed treecreeper. The NDVI is not only positively correlated to net primary productivity (*Myneni et al., 1995*; *Pettorelli et al., 2005*), but also to the structural complexity of forests (*Manes et al., 2010*). As a consequence, among the same forest type, a higher NDVI is related, given that all other variables are comparable, to more structured, multi-layered forests or to forest patches that are more productive or that have a higher leaf area index, where specialist birds can find a more suitable habitat (*Newton, 1994*; *Carrillo-Rubio et al., 2014*). Obviously, this conclusion also depends on the patch size and the degree of fragmentation, which are intertwined with NDVI and forest type. Indeed, a substantial influence of landscape structure in defining habitat suitability was clearly apparent when taking into account the three metrics together. Responses to fragmentation are species-specific and, usually, the more specialist a species, the more negative its response (*Devictor, Julliard & Jiguet, 2008*; *Rueda et al., 2013*). SDM outputs showed higher HS in localities in less fragmented landscapes, in agreement with the literature on forest specialist birds (*Fahrig, 2003*).

We used hierarchical statistical analysis of abundance to obtain unbiased estimates, corrected for detectability (*Royle, 2004a*). The significant difference in abundances between Bosco Pennataro and Chiarano-Sparvera is matched by the difference in the suitability of the two forests. Therefore, differences in abundance, HS and landscape metrics matched the same pattern: in locations with more degraded forest, both HS and abundance scored lower values, even though abundance showed higher variability, confounding the hypothesised relationships with HS.

Our results suggest that there is a positive relationship between habitat suitability and treecreeper abundance, even if the hypothesised triangular envelope (*VanDerWal et al., 2009*) did not emerge. However, its predictive power was quite weak, due to high abundance variability in both low and high HS locations. Extensive research has yielded little evidence

for the relationship between demographic parameters and HS (*Pearce & Ferrier, 2001*; *Nielsen et al., 2005*; *Jiménez-Valverde et al., 2009*). Related findings are often discordant (*Jiménez-Valverde et al., 2009*; *Tôrres et al., 2012*) and many concerns are raised on the controversial and often unconfirmed empirical relationships between ecological processes and landscape patterns (*Turner, Gardner & O'Neill, 2001*; *Kupfer, 2012*). That said, the relationship can be masked by the many unmodelled environmental variables that can conceal local suitability (*Lobo, Jiménez-Valverde & Real, 2008*). For this reason, *VanDerWal et al. (2009)* concluded that just the upper limit of abundance, and not its mean value, is predictable from HS. However, this relationship has been widely found to be very weak due to the difficulty to obtain reliable estimates of both abundance and HS (*Jiménez-Valverde, 2011*; *Oliver et al., 2012*; *Tôrres et al., 2012*). Some exceptions are presumably due to the use of indexes of abundance, instead of actual estimates (*De Moraes Weber & Viveiros Grelle, 2012*; *Gutiérrez et al., 2013*). Indeed, our approach was based not only on abundance estimates but also on HS values from different algorithms and averaged over the likely home range size. Moreover, our use of landscape features as predictive variables could have enhanced model performance since other studies (e.g., *Tôrres et al., 2012*), based mostly on climatic variables, found positive but weaker relationships between HS and abundance.

This result, though confirming the existence of a relationship, also highlights the limits of the SDM approach, suggesting that low HS can also occur in areas of high abundance, probably due to environmental factors that are not considered in modelling which may increase the actual HS of the area.

## CONCLUSION

Birds are considered good biodiversity indicators, especially to monitor habitat alteration (e.g., fragmentation) (*Gregory et al., 2008*; *Carrillo-Rubio et al., 2014*; *Czeszczewik et al., 2014*). For instance, in the context of biotic homogenization, one likely effect is the disappearance of specialist species which are more closely associated to unaltered forests (*McKinney & Lockwood, 1999*). Negative effects of habitat alteration can persist over years (*Kendrick et al., 2014*). Thus identification of the main species-habitat relationships is important to prevent the disappearance of more susceptible species (*Villard, Trzcinski & Merriam, 1999*; *King & DeGraaf, 2000*). Further, fragmentation can cause the disappearance of the specialist component of biodiversity (*Fahrig, 2003*). Such processes can alter biological, ecological and demographic traits like brood survival and growth (*Suorsa et al., 2003*; *Le Tortorec et al., 2012*), occupancy or population size (*Schmiegelow, Machtans & Hannon, 1997*; *Villard, Trzcinski & Merriam, 1999*; *Cooper & Walters, 2002*). Through SDMs, such results can be transposed into geographic projection and inform conservationists and practitioners (*Ferrier et al., 2007*; *Maiorano et al., 2015*). Therefore, modelling how fragmentation can affect the distribution of a species and understand the eventual relations with population decrease, can greatly improve conservation and management plans.

A forest landscape is, in most European cases, a human-modified landscape whose properties, like patch size, can affect many species (*Gil-Tena, Torras & Saura, 2008*). Our approach takes into account such issues in order to provide information-based advice.

In this way, we define the relationships between a species and some "directly adjustable" landscape features. The Chiarano-Sparvera forest stand is naturally located in a more fragmented landscape than is Bosco Pennataro. Hence, the abundance response (i.e., decrease) of the short-toed treecreeper is matched by habitat choice. Fragmentation, extensive rejuvenation of forest stands and tree plantations are all factors that can contribute to alter the suitability of an area. Since habitat alteration can decrease species abundance sooner than effectively reducing their geographic range (*Shoo, Williams & Hero, 2005*), identification of areas of low HS, where impact on abundance is more likely to cause local extinctions, could act as an early warning for species conservation. In our approach, these threats can occur on a large scale, can be related to possible changes in abundance and then used to inform practitioners and managers. Moreover, prediction of future land use scenarios can be implemented.

However, our results are a case study, limited to a single specialist species, strictly linked to mature well-preserved forests. This approach could be extended over different kinds of habitats and species, other than forests. Moreover, the modelling should be refined to include other potential resources and limiting factors, whether biotic or abiotic, in order to obtain more robust HS prediction (*Guisan & Thuiller, 2005*). The magnitude of the relationship between HS and abundance can then be used as a form of model validation (*Lobo, Jiménez-Valverde & Real, 2008*), thus helping to steer sound land use management and conservation planning.

## ACKNOWLEDGEMENTS

We are grateful to Jorge Soberón for valuable advice on the manuscript. We are also grateful to an anonymous reviewer for good advices. We are grateful to the secretary of the project MITO2000, especially to Simonetta Cutini, who patiently organized part of the dataset. Thanks are also due to Andrea Mancinelli for having assisted in many ornithological surveys and, in general, to the staff of the Isernia and Castel di Sangro Ufficio Territoriale Biodiversità del Corpo Forestale dello Stato (National Forest Service) for their logistic support. We are most grateful to the association ARDEA (www.ardeaonlus.it) for supporting our field work with its large number of volunteers.

### Funding

This research is part of the Life+ ManFor C.BD. project and hence has been co funded by a LIFE09 ENV/IT/000078 grant. The funders had no role in study design, data collection and analysis, decision to publish, or preparation of the manuscript.

### Grant Disclosures

The following grant information was disclosed by the authors:
LIFE09 ENV/IT/000078.
## Competing Interests

Marco Basile and Rosario Balestrieri declare that they have no competing interest in publishing this work as affiliated to the MItO2000 project.

The remaining authors declare there are no competing interests.

## Author Contributions

- Marco Basile conceived and designed the experiments, performed the experiments, analyzed the data, wrote the paper, prepared figures and/or tables.
- Francesco Valerio and Mario Posillico analyzed the data, prepared figures and/or tables, reviewed drafts of the paper.
- Rosario Balestrieri conceived and designed the experiments, performed the experiments, reviewed drafts of the paper.
- Rodolfo Bucci performed the experiments.
- Tiziana Altea and Bruno De Cinti commented on the manuscript.
- Giorgio Matteucci reviewed drafts of the paper.

## Data Availability

Basile M, Valerio F, Balestrieri R, Posillico M, Bucci R, Altea T, De Cinti B, Matteucci G. 2016. Patchiness of forest landscape can predict species distribution better than abundance: the case of a forest-dwelling passerine, the short-toed treecreeper, in central Italy. Mendeley Data. v2.

http://dx.doi.org/10.17632/p2z6rtcf8j.2.

## Supplemental Information

Supplemental information for this article can be found online at http://dx.doi.org/10.7717/peerj.2398#supplemental-information.

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
