# Peer review of "Patchiness of forest landscape can predict species distribution better than abundance: the case of a forest-dwelling passerine, the short-toed treecreeper, in central Italy"

_PeerJ, doi:10.7717/peerj.2398_

## Round 0.1 · original submission · Major Revisions

Both reviewers think that the paper is original and interesting, but both of them have a list of suggestions and commentaries that should be addressed before the paper can be published. And I also have some additional comments:

1) While reviewer 1 has comments on the title of the paper, in particular about the abundance (that depends on your answer to her questions, see below), I consider that the title of the paper should be more specific, including both the name of the bird and the region where the study was made: this information would make the paper more attractive to the readers interested in bird ecology, not only people interested in distributions. The title could be changed perhaps to something like: “Patchiness of forest landscape can predict species’ distribution better than its abundance: the case of a specialist bird, the short-toed treecreeper, in central-southern Italy”

2) In the abstract you need to give more information. Include the Latin name of the species and its family. Include, for instance, the location of the study, the number of records used, how you estimated abundance, and the different algorithms evaluated and which was best.
Remove the “a” in “…we define a relationships”.
Explain what is ““directly adjustable” landscape features”.
And please, improve, explain and make clearer the last phrase.

3) Line 101 and following lines: To evaluate how general and useful for other species may be the findings of the study we need some information about the ecology of the species:
Why and how it is a specialist bird? Does it Specialize in some kind of trees, and specific type of forest, or its diet is very specific? Is it a migratory bird, either latitudinal or in altitude or does it change of forest along the year?
What is the total distribution of the species?
Also indicate the family of the bird.

4) Line 105: Define ES (but also check comments of the two reviewers on this terminology).

5) Line 117 to 118: I am concerned, as also seems to be Reviewer 1, of using just part of the total distribution of a species to predict its SDM. Please comment if it is valid and what problems could be found.

6) Line 119: Change “felled” to “fell”.

7) Line 224: What is “c-hat”?

8) Lines 219 and following: I am concerned that, at the end of the day, there are only two data points to evaluate if abundance is related to suitability.

9) Line 312: Change “Gil-tena” to “Gil-Tena”; line 331 change “Jiménez-valverde” to “Jiménez-Valverde”.

Table 1, heading: Change “Surface” to “Area”.
Explain, very briefly what is “Orno-ostryetum” as most non-European readers will have no idea what means.

Reviewer 1 expressed concerns about the abundance issue. I do not think the title should be changed, but make clear in the text and in the abstract the relationship between fragmentation, abundance and distribution.

I do not agree with the idea of this reviewer of changing the name of “SDM” as it will become very confusing for the readers, but I would like to see, as mentioned above, a good discussion on whether the models work well with data of only a limited and localized part of the total distribution of the species.

This reviewer made some mistakes in the lines she mentioned, but I think you can find the location of each comment in you manuscript.

In relation the her comment 6, I think she is talking about “SE” (not AE) and suitability, but see also comments on this issue by Reviewer 2, comment 1 and 4.

Reviewer 2, Dr. Jorge Soberón, made a series of technical detail comments, in particular about Maxent and ACU methods, that you should review and answer with care, one by one, even if not numbered, and include a number of references that you should read and cite in the relevant sections.

Reviewer 1 ·

Basic reporting

1.- I consider that the authors do not assess the relation between forest fragmentation and abundance. This imply that the assertion made in the title is not robust. The authors perform analyses using several algorithms, where type forest and NDVI were the best environmental predictors for the presence of species. In addition, the correlation between abundance and ES is unclear making hard to see the association between patchiness and abundance . Consider changing the title so it reflects the results found by the authors.
2.-SDMs are inferences to species level, which means that the input data are a sample of the species. In this case the analyses were performed to a local scale, without including all the geographical range of the specie. Therefore this cannot be considered a SDMs. Please, justify the use of SDMs or change the term for a more appropriate one.

Experimental design

3.- line 151. Please, justify the use of different geographical resolution in the variables.
4.-line 196. The authors should explain how they defined ES.
5.- line 202. The paragraph is confusing. It is necessary to clarify the term "population size”, and how it is estimated.

Validity of the findings

6.- It is necessary to standardize the terminology, in some instances the authors use "AE" and in others "suitability".
7.- line 241. The authors mention "Quantile regression showed a positive relationships between abundance and ES (fig. 3 and fig. S2)". In the figures, a clear correlation pattern is not observed, the authors should indicate p and r values for clarity.
8.- In different sections of the discussion and conclusions, the authors allude to a positive relationship between ES and abundance, however, given the results, this relationship is not statistically robust. For this reason these sections should be modified.
9.- line 264. The paragraph is confusing, should be revised.
10. line 302. The paragraph is confusing, should be revised.

·

Basic reporting

Review of MS # 10636 “Patchiness of forest landscape can predict species’ distribution better than its abundance”
Jorge Soberón, University of Kansas

This manuscript attempts to relate abundance of a bird specialist on forest conditions, modeled with a hierarchical point-process, on indices of “environmental suitability” modeled with a variety of algorithms that have been used to estimate specie’s distributions (SDMs). The manuscript is clear (with one important exception), and the question can be clearly defined. The results, however, may or not be valid, since they are predicated on assuming that “environmental suitability” is the output of SDMs, something that may or not be the case.
The authors simply assume that the outputs of: (i) and a variety of algorithms (which may be based on rather different mathematical assumptions) , (ii) using presence data in combination with background sampling, or true absences, and (iii) assessed using AUCs, represent in the same way “environmental suitability.” It is known that this is not the case, most clearly for the Maxent algorithm (Royle et al., 2012). In SDM one attempts to estimate the probability of occurrence of a species, given a particular environment P(Y=1|e). However not all the methods the authors use estimate this probability of occurrence. In the first place, the authors apparently do not realize that Maxent is a “background sample” method (lines 147-148), which in turn makes Maxent incapable of estimating rigorously P(Y=1|e) without further assumptions. No “background” method can (Pearce & Boyce, 2006), without assumptions about randomness and detectability. Rather the raw output of Maxent (the authors do not say what settings they are using in Maxent) estimates P(e|Y=1) (Merow et al., 2013) which is an index of similarity to environments where a species has been observed, not an index of suitability. If the authors were using the logistic output of Maxent, that attempts to be a probability of occurrence, an entire set of other problems would arise (Royle et al., 2012), but the manuscript is not explicit about this most important point. The only way to estimate rigorously a distribution is by the use of presences and true-absences. In the absence of true-absences Maxent estimates some indices of similarity, and transforming them to P(Y=1|e) is not easy. Random sampling and even detectability allow presence-only data to be used to estimate P(Y=1|e), if the right model is used, but not Maxent. With the kind of data the authors are using, true-random samples are unlikely, although they document indistinguishable probability of detection.
But even if the authors were rigorously estimating P(Y=1|e), this is not necessarily an index of environmental suitability, because presence is not determined only by suitability of environmental variables, but also by movements and interactions with other species (Grinnell, 1917), and by availability of suitable conditions (Jackson & Overpeck, 2000). The authors fail to discuss this point, assuming with no elaboration that outputs of algorithms measure suitability. Maybe yes, or maybe not, and the point needs discussion.
Finally, authors are using AUCs as a method to evaluate models. Despite their popularity, AUCs have a large number of problems (Lobo et al., 2007) when applied to SDM, mostly when true-absences are not used.
Despite all the above I found the work interesting and useful, and I believe the paper could be published if the authors are capable of rewriting doing the following:
1) Avoid the term “environmental suitability” and use explicitly what the algorithms model (see Franklin, (2009)).
2) Use a more appropriate method to assess the goodness of fit of algorithms (Pearson et al., 2007).
3) Be explicit about the settings of every method you use (specifically, in Maxent report whether the raw or logistic output is used, what features are allowed, whether clamping is permitted or not…)
4) Modify and rewrite the discussion and conclusions to be explicit under what conditions “environmental suitability” can be estimated from “probability of presence,” , or by “probability of similarity” as the raw Maxent, and discuss your results in terms of the empirical relationship you obtain between such probability and abundance, at least for some algorithms. I also suggest that you elaborate more on the negative results that Torres et al. (2012) obtained when trying to correlate abundance to the output of SDMs. Notice that I say “the output of the SDM”, not “environmental suitability”. This is my main point.

Franklin, J. (2009) Mapping Species Distributions. Spatial Inference and Prediction, edn. Cambridge University Press, Cambridge.
Grinnell, J. (1917) The niche-relationships of the California Thrasher. Auk, 34, 427-433.
Jackson, S. T. & Overpeck, J. T. (2000) Responses of plant populations and communities to environmental changes of the late Quaternary. Paleobiology, 26 (Supplement), 194-220.
Lobo, J. M., Jiménez-Valverde, A. & Real, R. (2007) AUC: a misleading measure of the performance of predictive distribution models. Global Ecology and Biogeography, 17, 145-151.
Merow, C., Smith, M. J. & Silander, J. (2013) A practical guide to MaxEnt for modeling species' distributions: waht it does, and why inputs and settings matter. Ecography, 36, 1058-1069.
Pearce, J. & Boyce, M. S. (2006) Modelling distribution and abundance with presence-only data. Journal of Applied Ecology, 43, 405-412.
Pearson, R. G., Nakamura, M., Peterson, A. T. & Raxworthy, C. (2007) Predicting species distributions from small numbers of occurrence records: A test case using cryptic geckos in Madagascar. Journal of Biogeography, 34, 102-117.
Royle, J. A., Chandler, R. B., Yackulic, C. & Nichols, J. D. (2012) Likelihood analysis of species occurrence probability from presence-only data for modelling species distributions. Methods in Ecology and Evolution, 3, 545-554.
Tôrres, N. M., De Marco, P., Santos, T., Silveira, L., De Almeida Jácomo, A. T. & Diniz-Filho, J. a. F. (2012) Can species distribution modelling provide estimates of population densities? A case study with jaguars in the Neotropics. Diversity and Distributions, 18, 615-627.

Experimental design

The research question is not well defined, because you equate the output of miscellaneous SDMs with "environmental suitability"

Validity of the findings

The empirical results are clear and interesting. The interpretation needs work.

---

## Round 0.2 · Minor Revisions

I want to thank the authors for their carefully reviewed version of the paper. The different questions and concerns of the two reviewers and also my points were answered and/or evaluated in this new version.

Reviewer number 2 also read the new version, and she is only concerned with the maps in Figure S2, and I agree with her that they are difficult to visualize as they are now, in a grey scale. In particular some of the maps, like the one of the CTA or for GBM basically look almost completely black, and little information can be glanced from them. I would recommend the authors to improve them to make the easier to understand and more useful by using a color scale. This reviewer also asks to include by each map the habitat suitability values, and maybe some other associated statistics.

I have some minor additional points that should be considered before finally accepting the paper.

As the final version to be printed of the article will not be checked for English by the journal, the authors should be very careful with the language. Thus I am asking you to check all the spelling again, including the references, tables and supplementary materials, as I noticed some problems, but as I am not a native speaker, I may have missed several others.

Line 51: Change “need” to “needs”.

Lines 146: Change “non-foreste” to “non-forested”.

Line 178: It needs to be rewritten, to something that “would reveal if there is a spatial autocorrelations”…

Lines 206 to 209 and Figure 1: Show in Fig. 1 where are the two forest stands used to evaluate the birds. This way we can see how representative seems to be each site of study.
Also, I would like to see these sites marked in Figures S2, to see visualize the suitability of each area according to each of the models, for instance.

Lines 257: “The importance of the three landscape metrics”: I find this phrase confusing, as actually there are 6 variables in Fig. 2, but in the previous line you discuss only 2 of them, but in no case there are 3... Please, review again this line and the section.

Line 300: “employed an high”, I think should be changed to “employed a high”.

Line 471: Change “e

Reviewer 1 ·

Basic reporting

The manuscript explore the relationship between habitat suitability and abundance. Besides, the authors verified if these factors are affected by fragmentation. It is an interesting approach and might have potential to yield valuable results. However, here are some minor corrections.

Experimental design

no comment

Validity of the findings

The figures contained in the Supplementary Material Fig S2 are unclear. Colour employed in the figures, make a difficult interpretation. Moreover, its necessary to indicate the habitat suitability values associated to the figures.

---

## Round 0.3 · accepted · Accept

I think the new version of the manuscript was carefully corrected. I specially appreciate the effort if the authors in checking all the English by a native speaker and in improving in the figures.

I believe this is an important contribution for understanding the factors that affect the distribution of the species, the problems related to its modeling and for the conservation biology and understanding the role of fragmentation.